# Mutation Analysis of Radioresistant Early-Stage Cervical Cancer

**DOI:** 10.3390/ijms23010051

**Published:** 2021-12-21

**Authors:** Tae Oike, Yoshihito Sekiguchi, Yuya Yoshimoto, Takahiro Oike, Ken Ando, Wenchao Gu, Yasushi Sasaki, Takashi Tokino, Akira Iwase, Tatsuya Ohno

**Affiliations:** 1Department of Obstetrics and Gynecology, Gunma University Graduate School of Medicine, 3-39-22, Showa-machi, Maebashi 371-8511, Gunma, Japan; taeoike@gunma-u.ac.jp (T.O.); akiwase@gunma-u.ac.jp (A.I.); 2Department of Radiation Oncology, Gunma University Graduate School of Medicine, 3-39-22, Showa-machi, Maebashi 371-8511, Gunma, Japan; m11201059@gunma-u.ac.jp (Y.S.); k.ando0906@gunma-u.ac.jp (K.A.); tohno@gunma-u.ac.jp (T.O.); 3Department of Radiation Oncology, School of Medicine, Fukushima Medical University, 1, Hikarigaoka, Fukushima 960-1295, Fukushima, Japan; yuya.yoshi@gmail.com; 4Gunma University Heavy Ion Medical Center, 3-39-22, Showa-machi, Maebashi 371-8511, Gunma, Japan; 5Department of Diagnostic Radiology and Nuclear Medicine, Gunma University Graduate School of Medicine, Maebashi 371-8511, Gunma, Japan; sunferrero@gmail.com; 6Center for Medical Education, Division of Biology, Department of Liberal Arts and Sciences, Sapporo Medical University, Chuo-ku, Sapporo 060-8556, Hokkaido, Japan; yasushi@sapmed.ac.jp; 7Medical Genome Sciences, Research Institute for Frontier Medicine, Sapporo Medical University School of Medicine, Chuo-ku, Sapporo 060-8556, Hokkaido, Japan; tokino@sapmed.ac.jp

**Keywords:** cervical cancer, early stage, radiotherapy, radioresistance, mutation, *KRAS*, *SMAD4*

## Abstract

Radiotherapy is a definitive treatment for early-stage cervical cancer; however, a subset of this disease recurs locally, necessitating establishment of predictive biomarkers and treatment strategies. To address this issue, we performed gene panel-based sequencing of 18 stage IB cervical cancers treated with definitive radiotherapy, including two cases of local recurrence, followed by in vitro and in silico analyses. Simultaneous mutations in *KRAS* and *SMAD4* (*KRAS*^mt^/*SMAD4*^mt^) were detected only in a local recurrence case, indicating potential association of this mutation signature with radioresistance. In isogenic cell-based experiments, a combination of activating *KRAS* mutation and SMAD4 deficiency led to X-ray resistance, whereas either of these factors alone did not. Analysis of genomic data from 55,308 cancers showed a significant trend toward co-occurrence of mutations in *KRAS* and *SMAD4*. Gene Set Enrichment Analysis of the Cancer Cell Line Encyclopedia dataset suggested upregulation of the pathways involved in epithelial mesenchymal transition and inflammatory responses in *KRAS*^mt^/*SMAD4*^mt^ cancer cells. Notably, irradiation with therapeutic carbon ions led to robust killing of X-ray-resistant *KRAS*^mt^/*SMAD4*^mt^ cancer cells. These data indicate that the *KRAS*^mt^/*SMAD4*^mt^ signature is a potential predictor of radioresistance, and that carbon ion radiotherapy is a potential option to treat early-stage cervical cancers with the *KRAS*^mt^/*SMAD4*^mt^ signature.

## 1. Introduction

Cervical cancer occurs in approximately 0.5 million women worldwide annually, and mortality from this type of tumor ranks fourth among all cancers [1]. Radiotherapy is among the standard treatments for cervical cancer, along with surgery [2]. Owing to technological advancements in image-guided adaptive brachytherapy, stage IB cervical cancer (based on the International Federation of Gynecology and Obstetrics [FIGO] 2009 staging system) is almost curable by radiotherapy; the 5 year local control rates for IB1 and IB2 tumors are 98% and 92%, respectively [3]. Personalization for the remainder of tumors that harbor intrinsic radioresistance is important; thus, the establishment of predictive biomarkers and treatment options for this disease subset is needed.

Predictive biomarkers for tumor radioresistance have been pursued over the decades. For example, intratumoral hypoxia is associated with worse prognosis of cervical cancer treated with radiotherapy [4]. On the other hand, the radiosensitivity index (RSI), an algorithm composed of mRNA expression levels of ten genes, predicts the tumor responsiveness to radiotherapy; the clinical utility of RSI was validated by multiple cohorts [5]. Nevertheless, both tests are of relatively low convenience in the clinic as the former and the latter requires a polarographic oxygen electrode and a fresh-frozen tumor specimen, respectively. From this perspective, there is a need to develop tests that have higher clinical utility. Recently, gene panel tests have become widespread in the clinic. These tests can be performed using the leftover of formalin-fixed paraffin-embedded (FFPE) specimens prepared routinely for pathological diagnosis. Somatic mutation profiles identified by this approach are used to guide prediction of tumor response to molecular-targeted drugs [6], indicating the potential applicability of gene panel testing to radiotherapy personalization; however, mutation profiles associated with intrinsic radioresistance of early-stage cervical cancer have not been fully elucidated. To address this issue, we analyzed the mutation profiles of stage IB cervical cancer treated with definitive radiotherapy using a gene panel. In addition, we performed in vitro and in silico analyses to assess the contribution of candidate mutation profiles to radioresistance and to estimate the underlying mechanisms, respectively.

## 2. Results

### 2.1. Mutation Screening of Stage IB Cervical Tumors

To determine the mutation profile characteristics of radioresistant early-stage cervical cancers, we performed target-capture sequencing of pre-treatment tumor samples obtained from 18 patients with stage IB cervical cancer treated with definitive radiotherapy (Table 1). Among the 18 patients, two developed local recurrence post-radiotherapy: a 43-year-old patient with stage IB2 squamous cell carcinoma that recurred 17 months post-radiotherapy, and a 64-year-old patient with stage IB1 adenocarcinoma that recurred 11 months post-radiotherapy (referred to hereafter as Case #1 and Case #2, respectively).

Follow-up period and age data are presented as median (range). Adeno, adenocarcinoma; FIGO, the International Federation of Gynecology and Obstetrics 2009; HPV, human papilloma virus; LN, lymph node; PALN, para-aortic lymph node; Squamous, squamous cell carcinoma.

The status of 36 driver/tumor suppressor genes recurrently mutated in cervical cancer was evaluated by gene panel-based sequencing (see Section 4.2 for details). As a result, a total of 44 somatic mutations were identified (Figure 1 and Appendix A). Mutations in *PIK3CA* were identified in 16% (3/18) of patients; all of these mutations were well-characterized activating mutations in the helical-domain. The prevalence of *PIK3CA* mutations in this study cohort was in line with that reported in previous landmark studies using whole exome sequencing to analyze tumor-blood pairs obtained from patients with cervical cancer [7,8], suggesting that the analytical pipeline employed in the present study is robust.

Case #1 harbored *MLH1* R659* and *FGFR2* R6P mutations (Table 2). *MLH1* R659* is a recurrent mutation in various cancers (Figure 2A) and was deduced to be oncogenic loss-of-function mutation, whereas *FGFR2* R6P was deduced to be benign (Table 2) [9,10]. Case #2 harbored *KRAS* G12S and *SMAD4* P356L mutations (Table 2). *KRAS* G12S is a hotspot oncogenic gain-of-function mutation (Figure 2B), whereas the biological effect of *SMAD4* P356L is unclear (Table 2), although this mutation is recurrent in various cancers (Figure 2C) and is predicted to be oncogenic [9,10]. We focused on the simultaneous mutations in *KRAS* and *SMAD4* (*KRAS*^mt^/*SMAD4*^mt^) as a candidate mutation signature associated with radioresistance because these two mutations were not observed in the 16 non-recurrent tumors, in contrast to the *MLH1* and *FGFR2* mutations, which were observed also in non-recurrent tumors (Figure 1).

### 2.2. Investigation of the Effects of the KRAS^mt^/SMAD4^mt^ Signature In Vitro

To investigate association of the *KRAS*^mt^*/SMAD4*^mt^ signature with radioresistance, we performed in vitro experiments using isogenic cell systems. Despite an intensive literature search, we were unable to identify a *KRAS^mt^/SMAD4^mt^* cervical cancer cell line, consistent with clinical sequencing findings that none of the 287 cervical cancers in the Cancer Genome Atlas (TCGA) PanCancer Atlas dataset harbored this signature [11]. Hence, we used a KRAS- and SMAD4-wild-type colorectal adenocarcinoma cell line, SW48, and its genetically engineered derivative carrying the activating *KRAS*^G12D^ allele. The biological consequence of *SMAD4* P356L was reproduced by SMAD4 suppression using siRNA because SMAD4 functions as a tumor suppressor (Figure 3A) [12]. Interestingly, neither the presence of *KRAS*^G12D^ nor SMAD4 suppression alone influenced the sensitivity of cells to X-rays (Figure 3B). Notably, however, the combination of *KRAS*^G12D^ and SMAD4 suppression led to significantly greater resistance to X-rays compared with the parental cells (*p* = 0.019) (Figure 3B). These data indicate that simultaneous mutations in *KRAS* and *SMAD4* contribute to cancer cell radioresistance.

### 2.3. Gene Set Enrichment Analysis of Tumors Carrying the KRAS^mt^/SMAD4^mt^ Signature

Analysis of 55,308 cancers registered in a public database revealed a significant trend toward co-occurrence of mutations in *KRAS* and *SMAD4* (*p* < 0.0001; see Section 4.4 for details), indicating a potential synergistic role of these mutations in cancer survival and progression (Figure 4A) [11]. Thus, to further explore the biological characteristics of cancers with the *KRAS*^mt^/*SMAD4*^mt^ signature, we performed Gene Set Enrichment Analysis (GSEA) [13,14] using the Cancer Cell Line Encyclopedia (CCLE) dataset [15]. We chose pancreatic cancer for analysis because, in CCLE, this cancer type contains the greatest number of cell lines with the *KRAS*^mt^/*SMAD4*^mt^ signature (data not shown). Genes involved in epithelial mesenchymal transition (EMT), interferon alpha and gamma responses, KRAS signaling pathways, apical junction, and xenobiotic metabolism were significantly upregulated in cell lines with the *KRAS*^mt^/*SMAD4*^mt^ signature compared with those without; further, genes involved in the G2M checkpoint, MTORC1 signaling pathways, and cholesterol homeostasis, as well as transcriptional targets of MYC and E2F, were significantly downregulated (Figure 4B–D). These data indicate that enhancement of the EMT phenotype, inflammatory responses, and cell cycle are potential mechanisms underlying the radioresistance of cancers harboring simultaneous mutations in *KRAS* and *SMAD4*.

### 2.4. Sensitivity of Cells Carrying the KRAS^mt^/SMAD4^mt^ Signature to Carbon Ion Radiotherapy

Finally, to investigate possible treatment options for radioresistant *KRAS*^mt^/*SMAD4*^mt^ cancers, we evaluated the sensitivity of *KRAS*^mt^/*SMAD4*^mt^ cancer cells to carbon ions, as there is evidence that carbon ion radiotherapy has strong anti-tumor effects on photon resistant tumors [16]. Notably, in isogenic cell systems, irradiation with therapeutic carbon ion beams decreased the clonogenic survival of *KRAS*^G12D^-positive SMAD4-deficient cells to levels comparable to those of the parental cells (Figure 3C). These data indicate that carbon ion radiotherapy is a potential option for treatment of X-ray-resistant *KRAS*^mt^/*SMAD4*^mt^ cancers.

## 3. Discussion

We identified simultaneous mutations in *KRAS* and *SMAD4* in a pre-treatment tumor obtained from a case of stage IB cervical cancer with local recurrence. Contribution of the *KRAS*^mt^/*SMAD4*^mt^ signature to radioresistance was validated in vitro. To the best of our knowledge, this is the first report of a mutation profile associated with radioresistance in patients with early-stage cervical cancer who received definitive radiotherapy. In addition, our data demonstrate that the X-ray resistance of *KRAS*^mt^/*SMAD4*^mt^ cancer cells was overcome by therapeutic carbon ions. Although our findings are from a small cohort, the data indicate the possibility that the *KRAS*^mt^/*SMAD4*^mt^ signature is a potential predictor of radioresistance. This hypothesis is supported by a previous study reporting the presence of this mutational signature in a cervical tumor that recurred after radiotherapy [17].

The tumor from Case #2 was an HPV-negative adenocarcinoma, and both HPV-negativity and adenocarcinoma are predictors of worse prognosis [18,19]. At present, it is unclear whether simultaneous mutations in *KRAS* and *SMAD4* are enriched in HPV-negative tumors or adenocarcinoma, and collection of sufficient sample numbers to investigate this issue will be challenging because HPV-negative tumors and adenocarcinomas comprise minor subsets among cervical cancers. In our in silico analyses, the *KRAS*^mt^/*SMAD4*^mt^ signature was associated with upregulation of the EMT phenotype and inflammatory responses. Data from multiple studies suggest an association of the EMT phenotype with radioresistance, where resistance to apoptosis and enhanced DNA damage repair capacity are often described as the underlying mechanism [20,21,22,23]. By contrast, the influence of enhanced inflammatory responses on cancer cell radiosensitivity is inconclusive because this phenotype can either positively or negatively influence anti-tumor immunity [24], and warrants further research.

Our data also indicate that carbon ion radiotherapy is an option to treat radioresistant cervical cancers with the *KRAS*^mt^/*SMAD4*^mt^ signature. Carbon ion radiotherapy shows anti-tumor effects on cancers established as resistant to photon radiotherapy, such as pancreatic cancer, non-squamous cell head-and-neck tumors, and sarcomas [25,26,27]; however, it is a scarce treatment modality due to the limited number of facilities that are able to administer it worldwide [28]. Hence, with the findings from sequencing taken together, the clinical implication of the present data may be that, for early-stage cervical cancers considered for definitive radiotherapy, *KRAS*^mt^/*SMAD4*^mt^ signature can be tested pre-treatment using a routine FFPE specimen; then the positive (i.e., potentially radioresistant) cases can be stratified to carbon ion radiotherapy. This concept requires validation in a clinical setting.

The tumor from Case #1 had a putative oncogenic loss-of-function mutation in *MLH1*. *MLH1* is a DNA mismatch repair (MMR) gene; therefore, loss-of-function mutations in *MLH1* contribute to high levels of microsatellite instability (MSI-H) [29]. Approximately 3% of cervical cancers are deficient in MMR [30]. We did not pursue the influence of *MLH1* mutations on radiosensitivity in vitro and in silico in this study because the mutations were also observed in a case without local recurrence. Nevertheless, previous studies suggest the efficacy of PD-1 blockade on MMR-deficient MSI-H tumors [29,30]. From this perspective, the combination of PD-1 blockade with radiotherapy could be considered for patients with *MLH1*-mutated cervical cancers.

This study had the following limitations. First, the number of participants was small; however, this was inevitable because stage IB cervical cancers are predominantly treated with surgery. Indeed, no previous studies have analyzed the mutation profile of stage IB cervical cancer associated with local recurrence post-radiotherapy; therefore, the data presented herein are of value. In addition, we performed in vitro validation to complement the statistical weakness of the sequencing study. Second, matched normal tissue samples were lacking for a proportion of patients in the present cohort, due to challenges in the clinical workflow for sample collection. This lack of normal tissue samples may have negatively influenced the exclusion of single nucleotide polymorphisms (SNPs), although we used an established method for processing sequencing data in the absence of matched normal tissue samples [31]. Third, we sequenced a limited number of genes due to cost. Therefore, it is possible that mutations in genes that were not investigated in this study contributed to radioresistance. Lastly, we did not use normal cell lines in the siRNA experiments as control. Nevertheless, in the Human Protein Altas, there are no evident difference in SMAD4 expression between cancer cell lines and normal cell lines [32]. Based on this knowledge, we believe that the lack of data on normal cells does not severely diminish the value of the results of this experiment.

In summary, we identified simultaneous mutations in *KRAS* and *SMAD4* in a pre-treatment tumor obtained from a case of stage IB cervical cancer with local recurrence. *KRAS*^mt^/*SMAD4*^mt^ cancer cells showed resistance to X-rays in vitro, which was overcome by therapeutic carbon ions. These data indicate that the *KRAS*^mt^/*SMAD4*^mt^ signature is a potential predictor of radioresistance, and that carbon ion radiotherapy is an option to treat early-stage cervical cancers with the *KRAS*^mt^/*SMAD4*^mt^ signature, and warrants validation in a larger cohort.

## 4. Materials and Methods

### 4.1. Patients

Patients who met the following inclusion criteria were retrospectively enrolled in the present study: (i) newly diagnosed and pathologically confirmed cervical cancer; (ii) staged as IB, based on the FIGO 2009 staging system; (iii) treated with definitive radiotherapy [33] at Gunma University Hospital from 2006 to 2019; and (iv) DNA extracted from pre-treatment tumors available.

Post-treatment follow-up was performed as previously described [33]. Briefly, patients were followed-up every 1–3 months for the first 2 years post-radiotherapy, and then every 3–6 months for the subsequent 3 years (N.B., the first day of radiotherapy was defined as day 1). Disease status was assessed at each follow-up by gynecological examination and imaging (computed tomography or magnetic resonance). Local recurrence was defined as regrowth of primary tumor.

This study was conducted in accordance with the principles of the Declaration of Helsinki and approved by the Institutional Ethical Review Committee of Gunma University Hospital (approval number, 1109). Acquisition of informed consent was waived by the Institutional Ethical Review Committee, due to the opt-out design of the study, for those who enrolled in 2006–2013, whereas written informed consent was obtained from all participants enrolled in 2014–2019.

### 4.2. Identification of Somatic Mutations

Somatic mutations in pre-treatment tumors were analyzed using a commercially available Ion AmpliSeq Comprehensive Cancer Panel (CCP; Thermo Fisher Scientific, Waltham, MA, USA) to screen 409 oncogenes/tumor suppressor genes or a custom-made gene panel (43 genes selected based on the CCP), and 36 genes frequently mutated in cervical cancer [31], common to both panels (Appendix A).

Target-capture sequencing was performed as previously described [31]. Briefly, DNA was extracted from FFPE tumor tissues using the QIAamp DNA FFPE Tissue kit (Qiagen, Hilden, Germany). DNA fragmentation was examined using a TaqMan RNase P Detection Reagents kit and the FFPE DNA QC Assay (Thermo Fisher Scientific). Amplicon libraries were prepared using the Ion AmpliSeq Library Kit 2.0 (Thermo Fisher Scientific). Nucleotide sequencing was conducted using an Ion Torrent sequencer (Thermo Fisher Scientific). Sequence data were analyzed using Ion Torrent systems (Thermo Fisher Scientific) with the Genome Reference Consortium Human Build 37 (hg19) as the reference sequence. SNPs also present in sequencing data from matched blood samples or subject NA12878 from the 1000 Genomes project were removed. Somatic mutations were identified using previously reported criteria [31].

### 4.3. Human Papillomavirus Genotyping

Genotypes of human papillomavirus (HPV; type 6, 11, 16, 18, 30, 31, 33, 35, 39, 45, 51, 52, 56, 58, 59, and 66) were determined using the PapiPlex PCR method (GeneticLab Co., Ltd., Sapporo, Japan).

### 4.4. Mutation Analysis of Data from a Public Database

The biological effects of given mutations were estimated using OncoKB [9] and ClinVar [10]. The prevalence and location of given mutations were analyzed using cBioportal [11]. The datasets “Curated set of non-redundant studies” and “TCGA PanCancer Atlas” were used to analyze mutations in all types of cancer and those in cervical cancer, respectively.

### 4.5. Cell Culture

SW48 and SW48 KRAS (G12D/+) were obtained from ATCC (Manassas, VA, USA). Cells were cultured in RPMI-1640 (Sigma-Aldrich, St. Louis, MO, USA) supplemented with 10% fetal bovine serum (Life Technologies, Carlsbad, CA, USA) at 37 °C with 5% CO_2_.

### 4.6. siRNA Knockdown

siRNA transfection was performed using HiPerFect (Qiagen), as previously described [34]. Briefly, siSMAD4 (s8405, Thermo Fisher Scientific) or Control-siRNA (GGGAUACCUAGACGUUCUAdTdT, Sigma-Aldrich) was added to suspended cells after trypsinization. After 24 h, cells resuspended after trypsinization were re-transfected with siRNAs. Cells were incubated for 24 h after the second transfection, and then subjected to immunoblotting or clonogenic assays in parallel.

### 4.7. Immunoblotting

Immunoblotting was performed using whole cell lysates, as previously described [35]. Band intensities for proteins of interest were quantified using ImageJ v1.48 (National Institutes of Health, Bethesda, MD, USA) and normalized to that for GAPDH, a loading control. SMAD (438,454) and GAPDH (3638) antibodies (both from Cell Signaling Technology, Danvers, MA, USA) were used. The uncropped versions of immunoblot images are shown as Appendix A.

### 4.8. Irradiation

X-ray irradiation (160 kVp, 1.06 Gy/min) was administered using an MX-160Labo instrument (mediXtec, Matsudo, Japan). Carbon ion irradiation was performed at Gunma University Heavy Ion Medical Center. Cells were irradiated with carbon ions using a clinically-relevant setting (i.e., energy of 290 MeV/nucleon at the center of a 6 cm-spread-out Bragg peak; the dose-averaged linear energy transfer was approximately 50  keV/μm).

### 4.9. Clonogenic Assays

Clonogenic assays were performed using pre-irradiation plating methods, as previously described [36,37]. Colonies comprising at least 50 cells were counted. Surviving fraction values were normalized to those of corresponding controls. SF_2_ (i.e., clonogenic survival after 2 Gy irradiation) was used as the endpoint for X-ray sensitivity [38]. SF_1_ (i.e., clonogenic survival after 1 Gy irradiation) was used as the endpoint for carbon ion sensitivity, based on the fact that the relative biological effectiveness of carbon ions over photons is approximately 2 [39,40,41].

### 4.10. Gene Set Enrichment Analysis

For the pancreatic cancer cell lines registered to CCLE [15], *KRAS* and *SMAD4* mutation status were determined based on OncoKB [9]; mutations indicated as ‘(likely) oncogenic’ and ‘(likely) gain-of-function’ were considered to be positive for *KRAS*, whereas those indicated as ‘(likely) oncogenic’ and ‘(likely) loss-of-function’ were considered to be positive for *SMAD4* (Appendix A). RNAseq gene expression data from 1019 cell lines were download from CCLE [42]. Differentially expressed genes (DEGs) between *KRAS*^mt^/*SMAD4*^mt^ cell lines and others were determined using the R package, ‘limma’ [43]. Then, enrichment of DEGs in given biological signaling pathways, registered to the GSEA Hallmark gene set (i.e., h.all.v7.2.symbols) [44], was assessed using the R package, ‘clusterProfiler’ [14].

### 4.11. Statistical Analysis

Differences in clonogenic survival between two groups were examined by Mann–Whitney U test. Trends in a contingency table were examined by Chi-squared test. Statistical analyses were performed using GraphPad Prism 8 (GraphPad Software, Inc., San Diego, CA, USA). GSEA was performed using R (R Foundation for Statistical Computing, Vienna, Austria), where the level of statistical significance was determined based on previously described criteria [45].

## Figures and Tables

**Figure 1 ijms-23-00051-f001:**
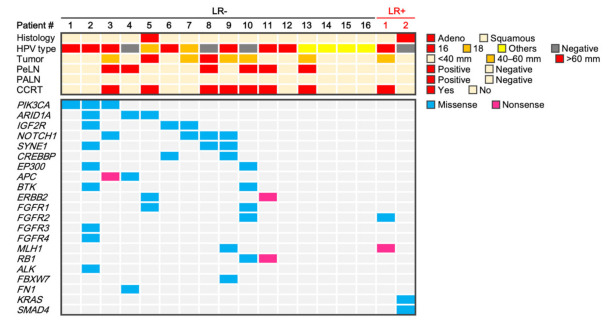
Summary of clinical factors and somatic mutations for 18 stage IB cervical cancers treated with radiotherapy. Adeno, adenocarcinoma; HPV, human papilloma virus; LR, local recurrence; PeLN, pelvic lymph nodes; PALN, para-aortic lymph nodes; Squamous, squamous cell carcinoma.

**Figure 2 ijms-23-00051-f002:**
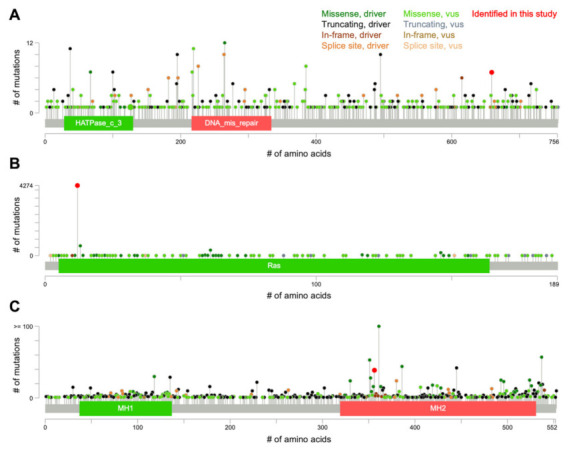
Somatic mutation spectra for (**A**) *MLH1*, (**B**) *KRAS*, and (**C**) *SMAD4* in 56,990 cancers of various origins. Figures were created on cBioPortal using the dataset, ‘curated set of non-redundant studies’, with modifications. vus, variant of unknown significance, #, number.

**Figure 3 ijms-23-00051-f003:**
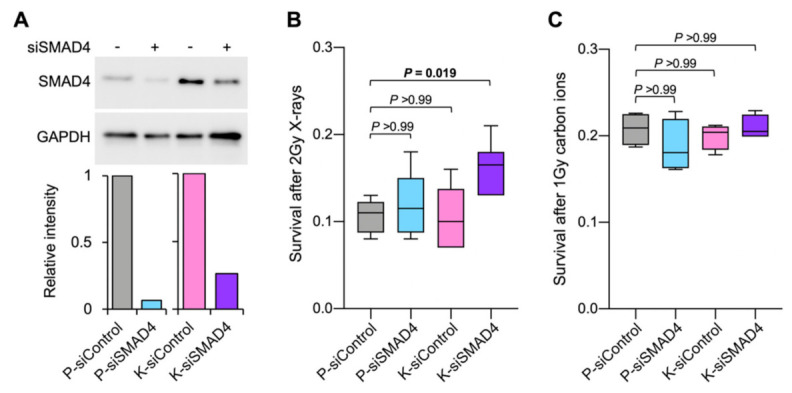
In vitro association of *KRAS* and *SMAD4* mutations with cancer cell sensitivity to X-rays or carbon ions. (**A**) Immunoblots showing SMAD4 suppression by siRNA treatment. Band intensities were quantified using ImageJ v1.48, and data shown are normalized to those for GAPDH, a loading control. (**B**) Clonogenic survival after 2 Gy X-rays (*n* = 6). (**C**) Clonogenic survival after 1 Gy carbon ions (*n* = 4). Surviving fractions shown are after normalization to those of unirradiated controls for each setting. *p* values shown were calculated by Mann–Whitney U test followed by Bonferroni correction. P, parental SW48 cells; K, SW48 cells carrying a *KRAS*^G12D^ allele.

**Figure 4 ijms-23-00051-f004:**
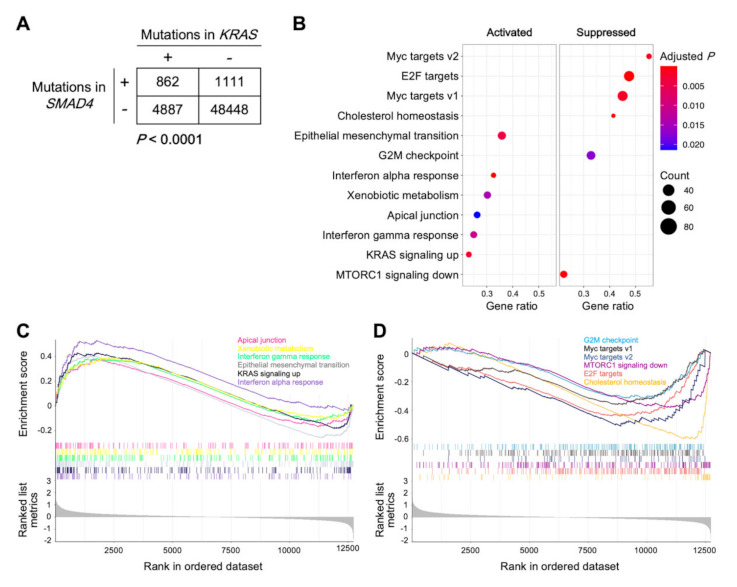
Analysis of the biological effect of *KRAS* and *SMAD4* mutations. (**A**) *KRAS* and *SMAD4* mutation status in 55,308 cancers registered in the cBioPortal, ‘curated set of non-redundant studies’. *p* values shown were calculated by Chi-squared test. (**B**–**D**) Gene Set Enrichment Analysis of biological pathways up- or downregulated in *KRAS*^mt^/*SMAD4*^mt^ pancreatic cancer cell lines (*n* = 11) compared with the other pancreatic cancer cell lines (*n* = 31) registered to the Cancer Cell Line Encyclopedia. **B**, dot plots. **C** and **D**, enrichment plots for up- and downregulated pathways, respectively.

**Table 1 ijms-23-00051-t001:** Patient characteristics.

Characteristic	All (*n* = 18)	Case #1	Case #2
Follow-up period (months)	63 (19–124)	23	19
Age (years)	15 (27–80)	43	64
Histological type			
Squamous	16	1	
Adeno	2		1
HPV type			
16	8	1	
18	2		
Others	4		
Not detected	4		1
FIGO			
IB1	10		1
IB2	8	1	
Tumor diameter (mm)			
<40	10		1
40–60	6	1	
>60	2		
Pelvic LN involvement			
Negative	12	1	1
Positive	6		
PALN involvement			
Negative	18	1	1
Positive	0		
Concurrent chemotherapy			
Yes	8	1	
No	10		1

**Table 2 ijms-23-00051-t002:** Prediction of the biological effects of mutations identified in cases with local recurrence.

Patient	Gene	Mutation	OncoKB		ClinVar
			Oncogenicity	Function	
Case #1	*MLH1*	R659*	Likely oncogenic	Likely LoF	Pathogenic
	*FGFR2*	R6P	Unknown	Unknown	Benign
Case #2	*KRAS*	G12S	Oncogenic	GoF	Pathogenic
	*SMAD4*	P356L	Predicted oncogenic	Unknown	Not listed

LoF, loss-of-function; GoF, gain-of-function. OncoKB [9]. ClinVar [10].

## Data Availability

Data presented in this paper is not available based on approved by the Institutional Ethical Review Committee of Gunma University Hospital (approval number, 1109).

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
