# Peer review of "Mutation Analysis of Radioresistant Early-Stage Cervical Cancer"

_ijms, 2021, doi:10.3390/ijms23010051_

Round 1

Reviewer 1 Report

The present paper cannot be accepted as original paper, cosidering the sequencing done only on two samples. This should be considered as case report, considering the comparation of the mutational patter of this two with public data sets and a siRNA validation on cell lines.

For siRNA study on SMAD4, no assement on the expression level was not assesed on the cell lines versus a normal cell line.

Author Response

Reviewer 1

The present paper cannot be accepted as original paper, considering the sequencing done only on two samples. This should be considered as case report, considering the comparation of the mutational patter of this two with public data sets and a siRNA validation on cell lines.

Response:

We thank the reviewer for evaluating our manuscript.

For siRNA study on SMAD4, no assessment on the expression level was not assessed on the cell lines versus a normal cell line.

Response:

We sincerely thank the reviewer for the important comment. In the Human Protein Altas, we do not find obvious difference in SMAD4 expression between cancer cell lines and normal cell lines (https://www.proteinatlas.org). Therefore, we believe that the lack of data on normal cells does not severely diminish the value of the results of this experiment. This was described as the limitation of this study (lines 226230 and Reference #32).

Reviewer 2 Report

I found revised paper well-designed and clearly presented. Clinical problem is maybe not very common, however it is worth to be studied due to lack of data. It may increase our success rate in some oncological patients. The authors defined weak points/limitations of the study. The methodology is proper, references are up to date. I would recommend publication of submitted paper in its present form. I would omit appendix figure a1, it may be visible in supplementary materials however unnecessarily. 

Author Response

Reviewer 2

I found revised paper well-designed and clearly presented. Clinical problem is maybe not very common, however it is worth to be studied due to lack of data. It may increase our success rate in some oncological patients. The authors defined weak points/limitations of the study. The methodology is proper, references are up to date. I would recommend publication of submitted paper in its present form. I would omit appendix figure a1, it may be visible in supplementary materials however unnecessarily.

Response:

We sincerely thank the reviewer for evaluating our manuscript and for the encouraging comments. According to the suggestion, Appendix Figure A1 was omitted and transferred to Supplementary Data S3. Corresponding parts in the manuscript was revised accordingly (lines 301 and 336).

Reviewer 3 Report

I found revised paper well-designed and clearly presented. Clinical problem is maybe not very common, however it is worth to be studied due to lack of data. It may increase our success rate in some oncological patients. The authors defined weak points/limitations of the study. The methodology is proper, references are up to date. I would recommend publication of submitted paper in its present form. 

Author Response

Reviewer 3

I found revised paper well-designed and clearly presented. Clinical problem is maybe not very common, however it is worth to be studied due to lack of data. It may increase our success rate in some oncological patients. The authors defined weak points/limitations of the study. The methodology is proper, references are up to date. I would recommend publication of submitted paper in its present form.

Response:

We sincerely thank the reviewer for evaluating our manuscript and for the encouraging comments.

Reviewer 4 Report

Despite the well-developed cervical cancer screening, the disease has growing incidence and prevalence especially in the developing countries.  In a light of the WHO strategy to eliminate cervical cancer by 2050, the manuscript presented for the review covers a very important part of medicine trying to develop precision approach to CC patients treatments.

The study is well-designed, the text in general is very well written. The references are up-to-date. However, some corrections should be made before acceptance and  publishing.

I would suggest expanding a little more the introduction part to give a better understanding to potential readers.

The methods part is detailed enough. The result section is clear and supplies with tables and figures sufficiently.

The discussion part is well-structured. It could be improved by adding a clinical implication of the study results.

Appropriate references are picked for the manuscript.

Author Response

Reviewer 4

Despite the well-developed cervical cancer screening, the disease has growing incidence and prevalence especially in the developing countries. In a light of the WHO strategy to eliminate cervical cancer by 2050, the manuscript presented for the review covers a very important part of medicine trying to develop precision approach to CC patients treatments. The study is well-designed, the text in general is very well written. The references are up-to-date. However, some corrections should be made before acceptance and publishing.

Response:

We sincerely thank the reviewer for evaluating our manuscript and for the encouraging comments. According to the suggestions, we made a thorough revision on our manuscript as follows.

I would suggest expanding a little more the introduction part to give a better understanding to potential readers.

Response:

We sincerely thank the reviewer for the important comment. According to the suggestion, we added the following sentences that acknowledge previous studies on the prediction of tumor radioresistance: "Predictive biomarkers for tumor radioresistance have been pursued over the decades. For example, intratumoral hypoxia is associated with worse prognosis of cervical cancer treated with radiotherapy (Suzuki et al. Int J Gynecol Cancer 2006;16:306–311). On the other hand, the radiosensitivity index (RSI), an algorithm composed of mRNA expression levels of ten genes, predicts the tumor responsiveness to radiotherapy; the clinical utility of RSI was validated by multiple cohorts (Torres-Roca. Per Med 2012;9:547–557). Nevertheless, both tests are of relatively low convenience in the clinic as the former and the latter requires a polarographic oxygen electrode and a fresh-frozen tumor specimen, respectively. From this perspective, there is a need to develop tests that have higher clinical utility. Recently, gene panel tests have become widespread in the clinic. These tests can be performed using the leftover of formalin-fixed paraffin-embedded (FFPE) specimens prepared routinely for pathological diagnosis (lines 4960)". The references #4 and #5 were added accordingly.

It could be improved by adding a clinical implication of the study results.

Response:

We sincerely thank the reviewer for the important comment. According to the suggestion, we added the following sentences to explain clinical implication of the study results: "clinical implication of the present data may be that, for early-stage cervical cancers considered for definitive radiotherapy, KRASmt/SMAD4mt signature can be tested pre-treatment using a routine FFPE specimen; then the positive (i.e., potentially radioresistant) cases can be stratified to carbon ion radiotherapy. (lines 200203)".

The methods part is detailed enough. The result section is clear and supplies with tables and figures sufficiently. The discussion part is well-structured. Appropriate references are picked for the manuscript.

Response:

We thank the reviewer for the comments.
